# The Influence of Ambient Temperature Changes on the Indicators of Inflammation and Oxidative Damage in Blood after Submaximal Exercise

**DOI:** 10.3390/antiox11122445

**Published:** 2022-12-12

**Authors:** Marta Pawłowska, Celestyna Mila-Kierzenkowska, Tomasz Boraczyński, Michał Boraczyński, Karolina Szewczyk-Golec, Paweł Sutkowy, Roland Wesołowski, Marlena Budek, Alina Woźniak

**Affiliations:** 1Department of Medical Biology and Biochemistry, Ludwik Rydygier Collegium Medicum in Bydgoszcz, Nicolaus Copernicus University in Toruń, 87-100 Toruń, Poland; 2Department of Health Sciences, Olsztyn University College, 10-283 Olsztyn, Poland; 3Department of Health Sciences, Collegium Medicum, University of Warmia and Mazury in Olsztyn, 10-719 Olsztyn, Poland

**Keywords:** exercise, inflammation, cytokines, lysosomal enzymes, serine protease inhibitor, oxidative stress, cold water immersion, sauna bath

## Abstract

Physical activity has a positive effect on human health and well-being, but intense exercise can cause adverse changes in the organism, leading to the development of oxidative stress and inflammation. The aim of the study was to determine the effect of short-term cold water immersion (CWI) and a sauna bath as methods of postexercise regeneration on the indicators of inflammation and oxidative damage in the blood of healthy recreational athletes. Forty-five male volunteers divided into two groups: ‘winter swimmers’ who regularly use winter baths (*n* = 22, average age 43.2 ± 5.9 years) and ‘novices’ who had not used winter baths regularly before (*n* = 23, mean age 25 ± 4.8 years) participated in the study. The research was divided into two experiments, differing in the method of postexercise regeneration used, CWI (Experiment I) and a sauna bath (Experiment II). During Experiment I, the volunteers were subjected to a 30-min aerobic exercise, combined with a 20-min rest at room temperature (RT-REST) or a 20-min rest at room temperature with an initial 3-min 8 °C water bath (CWI-REST). During the Experiment II, the volunteers were subjected to the same aerobic exercise, followed by a RT-REST or a sauna bath (SAUNA-REST). The blood samples were taken before physical exercise (control), immediately after exercise and 20 min after completion of regeneration. The concentrations of selected indicators of inflammation, including interleukin 1β (IL-1β), interleukin 6 (IL-6), interleukin 8 (IL-8), interleukin 8 (IL-8), interleukin 10 (IL-10), transforming growth factor β1 (TGF-β1) and tumor necrosis factor α (TNF-α), as well as the activity of indicators of oxidative damage: α1-antitrypsin (AAT) and lysosomal enzymes, including arylsulfatase A (ASA), acid phosphatase (AcP) and cathepsin D (CTS D), were determined. CWI seems to be a more effective post-exercise regeneration method to reduce the inflammatory response compared to a sauna bath. A single sauna bath is associated with the risk of proteolytic tissue damage, but disturbances of cellular homeostasis are less pronounced in people who regularly use cold water baths than in those who are not adapted to thermal stress.

## 1. Introduction

In recent years, there has been observed a growing interest in a healthy lifestyle. Undoubtedly, physical activity is widely recognized as an important part of healthy way of life. However, intense exercise can cause fatigue in the musculoskeletal and nervous systems and also result in adverse changes in the organism, leading to the development of inflammation [1]. Exercise-induced muscle injury (EIMD) is a common consequence of intense training. The recovery of muscles to their functional state before exercise sometimes takes up to several days [2]. The activation of inflammatory processes is essential for muscle repair and adaptation to exercise. On the other hand, excessive or persistent inflammation is thought to contribute to further tissue damage due to the nonspecific phagocytic function of inflammatory cells, such as neutrophils [3]. Oxidative stress is also an important element modulating the inflammatory response after exercise. The increasing demand for energy after exercise causes greater oxygen consumption by the mitochondria, which increases the formation of reactive oxygen species (ROS) [4]. High levels of ROS can cause the peroxidation of the cell membrane and destabilization of the structure of muscle cells [5,6]. Increased production of ROS can also lead to the disruption of lysosomal membrane integrity by peroxidation of lysosomal membrane lipids [7]. Structural changes in muscle fibers are also accompanied by increased release of certain intracellular enzymes, including those belonging to lysosomal acid hydrolases [8]. Thus, the released lysosomal enzymes can be used as markers of oxidative damage [9,10]. The mediators of the initiation of the inflammatory reaction are released at the site of damage to the muscle tissue as a result of the damage caused by ROS [5,11]. On the other hand, increased ROS levels during exercise can also contribute to some of the post-exercise adaptations [12]. Moderate exposure to ROS is necessary to trigger the organism’s adaptive responses, such as activation of antioxidant defenses [13].

Exercise can elicit an acute-phase response, characterized by an increase in peripheral blood levels of cytokines and chemokines [4]. Accumulation of pro-inflammatory cytokines, neutrophils and macrophages is responsible for the damage to and dysfunction of the muscle tissue [14]. Physiotherapeutic treatments, such as massage, kinesiotherapy and various methods that use low and high temperatures can be an effective way to support the process of rehabilitation of even minor injuries and can stimulate biological regeneration [15,16,17,18].

In recent years, the results of numerous studies have emphasized the positive effect of winter baths on the human organism [19,20,21]. Pain and anti-inflammatory therapies based on the use of low temperatures have long been used to treat many diseases [22,23]. However, exposure of the organism to low ambient temperatures can also cause physiological stress [9]. Cold water immersion (CWI) is believed to reduce blood cytokine levels, which may inhibit the development of inflammation after exercise [24]. On the other hand, it is known that low temperatures affect the mobilization of leukocytes [25]. In addition, it has been suggested that exposure to cold initiates changes in cytokine expression associated with a nonspecific acute phase response that may be the result of multiple interactions between cytokines and neuroendocrine hormones [26]. CWI is the process of immersing a part of or the whole body (except the head) in water with a temperature below 15 °C [27]. CWI is becoming more and more popular as a regenerative method for athletes due to its effectiveness and low costs [3,24,28]. Recent studies have begun to question the effectiveness of muscle cooling, providing many indications that neither inflammation nor muscle damage is actually reduced by CWI [29]. Still, the use of low temperatures remains the leading method of regeneration among competitive athletes after intense exercise. Many people justify using CWI after exercise based on its clear benefits [29]. The differences in the results of studies describing the effect of CWI on post-exercise regeneration of the organism may be related to the lack of standardization and a significant variety of CWI protocols (e.g., athlete’s level of training, type of training performed, exercise intensity, immersion time and water temperature) [30].

Exposition to a high ambient temperature during a sauna bath is an another form of thermotherapy. Increased organism temperature triggers a series of thermoregulatory reactions that prevent overheating [31]. It may lead to a reduction in the amount of ROS, which results in a reduction in oxidative stress and in the activity of the inflammatory pathway [32]. There is growing evidence that exposure to high ambient temperatures can trigger adaptive mechanisms such as the regulation of genes involved in muscle growth and differentiation [33]. On the other hand, dehydration and hyperthermia belong to the major dangers of excessive thermal stress [34]. In addition, being overweight and a sedentary lifestyle can further aggravate thermal stress [35,36]. There is little evidence for the effects of sauna baths on an athlete’s organism, but some studies have found that entering the sauna immediately after exercising provides an additional training stimulus [32,37]. The use of high temperatures is believed to be a useful treatment to reduce post-exercise soreness by increasing blood flow to target muscles, improving their oxygenation and helping to remove harmful metabolites produced during exercise from regenerating muscles [38].

Both CWI and sauna are supposed to be treatments that reduce inflammation in skeletal muscles. However, there are few scientific reports about the effect of exposure to cold or heat on pro-and anti-inflammatory cytokine levels. Moreover, most of the studies were performed in the population of professional athletes, while there are no studies on the general population and recreational athletes [39]. In addition, only few studies focused on the effect of CWI or post-exercise sauna treatment on markers of inflammation in less than two hours. This is why the aim of this study was to determine the effect of CWI and sauna baths on the concentration of selected cytokines and the activity of selected lysosomal enzymes in people regularly or sporadically exposed to cold and heat.

## 2. Materials and Methods

### 2.1. Participants

Forty-five healthy men participated in the study. The participants were divided into two groups: winter swimmers, people regularly undergoing winter baths (average age 43.2 ± 5.9 years) and novices who had not used winter baths regularly before (mean age 25 ± 4.8 years). The subjects did not change their eating habits (first of all did not take dietary supplements with antioxidant and anti-inflammatory activity) and physical activity, and did not undergo any therapy that could affect the parameters of inflammation in the organism, immediately before or during the study. The study was approved by the Bioethics Committee at Collegium Medicum in Bydgoszcz, UMK in Toruń, Poland (no. KB 278/2016). The participants were informed about the purpose of the study and the potential associated risks and gave their written consent to participate in the study.

A 170 Physical Fitness Test (PWC170) was performed to determine the participants’ aerobic fitness. PWC170 consisted of two 5-min standard exercise sessions on a bicycle ergometer (Monark Ergomedic 828 E). The load of the second exercise test was increased to obtain a heart rate (HR) of 170 beats per minute (bpm) but not to exceed it. The PWC170 index was calculated from the mean HR value recorded at the end of each 5-min exercise. HR was measured with a frequency meter (Polar Electro Oy, Finland). Load (power in watts, W) was calculated during exercise with a HR of 170 bpm. The test result was calculated from the formula [40]:PWC170 = P1 + (P2 − P1)/(170 − HR1) (HR2 − HR1)

P1—the power of the first exercise test;P2—the power of the second exercise test;HR1—HR during the first exercise test;HR2—HR during the second exercise test

The PWC170 value correlates well with the maximum oxygen consumption (VO_2max_), which is a fundamental indicator of the oxygen function [41]. VO_2max_ for all study participants during each stage of both experiments was calculated according to the Astrand-Ryhming nomogram using the PWC170 values [42]. Borg Category-Ratio-10 (CR10) scale with values ranging from “0” to “10” was used to assess the subjective level of fatigue (RPE). The first value “0” means “no effort” and the last “10” means “extremely hard” effort. There is also an effort index above 10, labeled as “∗”. It is the effort that makes the person “unable to continue” the exercise [43]. The RPE scale was used at each stage of both research experiments after 30 min of exercise.

In order to assess the level of daily physical activity, participants completed the International Physical Activity Questionnaire (IPAQ). In both study groups, the volunteers rated the level of weekly physical activity between “medium” and “high”. The energy cost of effort according to IPAQ is expressed in units called MET (metabolic equivalent of work). 1 MET is the value of your resting metabolic rate and is the volume of oxygen consumed in 1 min, which is approx. 3.5 mL/kg/min. Each type of physical activity can be expressed in units of MET-min/week by multiplying the coefficient assigned to this activity by the number of days of exercise in weeks and the duration in minutes per day [44]. The IPAQ level of “high” level means that the subjects met one of the following two criteria: three or more days of intense physical effort, a total of at least 1500 MET-min/week, seven or more days of any combination of efforts (walking, moderate or vigorous exercise) more than 3000 MET-min/week [44]. On the other hand, the “average” physical effort means that the subjects met one of the following three criteria: 3 or more days of intense physical activity, not less than 20 min a day; 5 or more days of moderate exercise or walking for no less than 30 min a day; 5 or more days of any combination of efforts (walking, moderate or vigorous exercise) exceeding 600 MET-min/week [44]. The basic characteristic of the studied groups is presented in Table 1.

The measurement of the body components (body fat, BF and total water content, TBW) of the participants was carried out using the Tanita bioelectric impedance analyzer—BC 418 MA (Tanita Corporation, Tokyo, Japan).

### 2.2. Study Design

The research was carried out at the Central Research Laboratory of the University of Olsztyn in Olsztyn, Poland and was divided into two experiments. The experiments differed in the method of post-workout regeneration used. In Experiment I it was an immersion in cold water, and in Experiment II it was a sauna treatment. The research was carried out in a cross-over study. Participants were assigned to control conditions (stage I) or to CWI/sauna treatment (stage II) in a cross-sequence (cross-over study). Each participant attended all stages (stage I and stage II during Experiment I, stage I and stage II during the Experiment II). The stages were separated by a week-long break. The scheme of the course of the research is presented in Figure 1.

During the first stage of Experiment I, the volunteers were subjected to 30-min aerobic exercise [with a power of 70% of the maximum heart rate—70% HRmax] on a bicycle ergometer [Monark Ergomedic 828 E]. HRmax was calculated according to the Oja, Tuxworth formula: HRmax = 205 − 0.5 × age [45]. Then, the participants rested in a sitting position at room temperature (passive regeneration, RT-REST). During the second stage (one week later), the participants performed an identical physical effort, but 5 min after its completion, they were subjected to a 3-min bath in a pool with cold water at 8 °C, and then rested in a sitting position for another minutes (up to 20 min). The participants of the experiment were dressed only in swimming trunks and they immersed the whole body in the water, except for the head and neck. During both sessions, blood samples were taken from each test subject three times from the median posterior vein: before exercise (control, BE), immediately after exercise (AE) and 20 min after each type of rest (RT-REST or CWI-REST). Experiment II was also divided into two stages. During the first stage, the subjects were subjected to the same exercise as in Experiment I on a bicycle ergometer [30 min, with a power of 70% of the maximum heart rate—70% HRmax]. Then the participants rested seated at room temperature (similar to Experiment I). During the second stage (one week later), the participants performed an identical exercise, but 5 min after its completion they underwent a sauna treatment. This treatment consisted of two 10-min entries to the sauna (temperature 85 °C, relative humidity 5–10%). After each exit from the sauna, the participants cooled the body in a shower with cold water and rested in a sitting position. The cooling and rest time was 3 min (in accordance with the generally recognized rules of the sauna). During both stages blood samples were taken from each examined person three times from the posterior vein: BE (control), AE and 20 min after rest, both RT-REST and the sauna bath (SAUNA-REST); the mean values of the trials were given as a result.

### 2.3. Determination of the Activity of α1-Antitripsin (AAT) and Lysosomal Enzymes

The determination of the activity of α1-antitrypsin in the blood serum was performed using the Eriksson method, the principle of which is to measure the decrease in the enzymatic activity of trypsin as a result of a short incubation with defibrinated blood serum [46]. The AAT activity is expressed in mg of inhibited trypsin/mL of serum. In order to determine the arylsulfatase A (ASA) activity, the Roy’s method modified by Błeszyński was used. The activity of this enzyme is measured as the amount of 4-nitrocatechol (4-NC) released during enzymatic hydrolysis of the substrate (4-nitrocatechol sulphate). ASA activity is expressed in nmol 4-NC/mg protein/min. Acid phosphatase activity was determined using the Bessy method in Krawczyński’s modification [47]. The activity of this enzyme is measured as the amount of p-nitrophenol released during enzymatic hydrolysis of the substrate, p-nitrophenol phosphate. The acid phosphatase activity is expressed in nmol p-nitrophenol/mg protein/min. Anson’s method was used to determine the activity of cathepsin D [42]. In this method, the addition of a phenolic reagent produces a blue color, the level of which is determined spectrophotometrically. Cathepsin D activity is expressed in nmoles of tyrosine/mg protein/min.

### 2.4. Determination of the Cytokine Concentrations

Ready-made analytical kits based on the ELISA method were used to determine the concentration of IL-1β, IL-6, IL-8, IL-10, TNF-α (expressed in pg/mL) and TGF-β1 (expressed in in ng/mL). Diaclone SAS (France) kits were used for the tests. The determinations were carried out in accordance with the instructions provided by the manufacturer. Microplates with wells coated with monoclonal antibodies highly specific for the cytokines detected were attached to each analytical kit.

### 2.5. Statystical Analysis

Statistical analysis was carried out using the STATISTICA 12 PL package. The obtained results were subjected to post-hoc statistical analysis (HSD Tukey’s test) using the one-way statistical test ANOVA (analysis of variance). When performing the analysis, all ANOVA assumptions regarding the equality among each group separately, homogeneity of variance (Levene’s test) and the assessment of compliance of the analyzed variables with the normal distribution (Kolmogorov–Smirnov test) were taken into account. The results are presented as the arithmetic mean value ± standard deviation (SD). Comparisons were considered as statistically significant when the calculated value of the tested probability was *p* < 0.05.

## 3. Results

### 3.1. The activity of α1-Antitripsin and Lysosomal Enzymes in Winter Swimmers

In both stages of Experiment I, no statistically significant changes in AAT and ASA and CTS D activity in the blood serum of winter swimmers after exercise (AE), as well as after exercise followed by the regeneration at room temperature (RT-REST), were found. Comparing the results obtained after RT-REST with the results obtained after the regeneration combined with CWI (CWI-REST), no statistically significant differences in AAT activity and the determined lysosomal enzymes (*p* > 0.05) were found (Figure 2a–d).

In the first stage of Experiment II (regeneration combined with sauna bath, SAUNA-REST), a statistically significant increase of about 46% in CTS D activity was observed AE versus BE (*p* < 0.05). Twenty minutes after RT-REST, the activity of this enzyme was approximately 52% lower than AE (*p* < 0.05) (Figure 3d). In the second stage of Experiment II (SAUNA-REST), a statistically significant increase of about 36% in AAT activity 20 min after SAUNA-REST, compared to BE (*p* < 0.001), was observed (Figure 3a). Moreover, it was shown that AAT activity 20 min after RT-REST was lower than 20 min after SAUNA-REST (*p* < 0.001) (Figure 3a). There were no statistically significant differences in the activity of lysosomal enzymes comparing RT-REST and SAUNA-REST.

Comparing the activity of AAT and lysosomal enzymes after CWI-REST with their activity after SAUNA-REST, statistically significant differences in the activity of AAT, AcP and ASA were observed. AAT activity 20 min after SAUNA-REST was approximately 55% higher than 20 min after CWI-REST (*p* < 0.01) (Figure 3a). Moreover, AcP activity was higher by about 52% (*p* < 0.01) (Figure 2b) and ASA activity by about 62% (*p* < 0.01) (Figure 3c) after SAUNA-REST than after CWI-REST.

### 3.2. The Activity of α1-Antitripsin and Lysosomal Enzymes in Novices

There were no statistically significant changes in the activity of the parameters tested in the blood serum of novices AE compared to BE (*p* > 0.05) in both stages of Experiment I (Figure 4a–d). Moreover, there were no statistically significant changes in the determined parameters after RT-REST and after CWI-REST versus BE and AE (*p* > 0.05) (Figure 4a–d).

In the second stage of Experiment II, the activity of AcP 20 min after SAUNA-REST was higher than AE (*p* < 0.001) and higher than BE (*p* < 0.001). Moreover, the activity of AcP 20 min after SAUNA-REST was higher than the activity of this enzyme 20 min after RT-REST (*p* < 0.001) (Figure 5b). The activity of CTS D AE was higher than BE (*p* < 0.001). However, 20 min after SAUNA-REST, the activity of this enzyme was lower than AE (*p* < 0.001) (Figure 5d).

Significantly lower CTS D activity in the blood serum of novices after SAUNA-REST compared to the activity of this enzyme after CWI-REST was observed (*p* < 0.05) (Figure 5d). Moreover, statistically significant higher activity of AcP after SAUNA-REST than after CWI-REST was noted (*p* < 0.05) (Figure 5b).

### 3.3. The Concentration of Cytokines in Winter Swimmers

In the first stage of Experiment I, a statistically significant increase in the concentration of IL-1β (*p* < 0.05) and IL-6 (*p* < 0.001) in the blood serum of winter swimmers AE compared to BE was observed (Figure 6a). However, 20 min after RT-REST, the concentration of IL-6 was statistically significantly lower than AE (*p* < 0.001) (Figure 6b). No statistically significant changes were observed in the concentration of IL-8, IL-10, TNF-α and TGF-β1 AE versus BE as well as after RT-REST (*p* > 0.05) (Figure 6c–f). In the second stage of Experiment I, IL-6 was almost 4-fold higher AE than control (*p* < 0.05). However, the concentration of IL-6 20 min after CWI-REST was lower than AE (*p* < 0.001) (Figure 6b). Moreover, a statistically significant increase in the concentration of IL-10 20 min after CWI-REST versus AE was found (*p* < 0.001). Twenty minutes after CWI-REST, the concentration of this cytokine was also more than 5-fold higher than BE (*p* < 0.05) (Figure 6d). However, no statistically significant changes in the concentration of IL-1β, IL-8, TNF-α and TGF-β1 AE and CWI-REST (*p* > 0.05) were observed (Figure 6a,c,e,f). A significantly lower concentration of IL-6 (*p* < 0.001) and a statistically significantly higher concentration of IL-10 (*p* < 0.001) after CWI-REST than after RT-REST were observed (Figure 6b,d).

In the first stage of Experiment II a statistically significant increase in the concentration of IL-1β of approx. 60% AE (*p* < 0.001) versus BE was noted. However, 20 min after RT-REST the concentration of this cytokine was lower by about 25% than AE (*p* < 0.001) (Figure 7a). There was also a two-fold increase in IL-6 concentration (*p* < 0.001) (Figure 7b) and a 3-fold increase in IL-10 concentration (*p* < 0.001) (Figure 7d) AE compared to BE. 20 min after RT-REST the concentration of IL-10 was lower than AE (*p* < 0.001), but was still twice as high as BE (*p* < 0.001). No statistically significant changes in the concentration of IL-8, TNF-α and TGF-β1 AE as well as after RT-REST were observed (*p* > 0.05) (Figure 7c,e,f). In the second stage of Experiment II, a statistically significant 6-fold increase in the concentration of IL-6 was observed AE (*p* < 0.001) versus BE. 20 min after SAUNA-REST, the concentration of this cytokine was statistically significantly lower than AE (*p* < 0.001) and still about three times higher than BE (*p* < 0.001) (Figure 7b). There was also a statistically significant increase in IL-8 concentration 20 min after SAUNA-REST versus AE (*p* < 0.05) (Figure 7c). Moreover, AE the concentration of IL-10 increased more than 2,5-fold versus BE (*p* < 0.001). In turn, 20 min after SAUNA-REST, the concentration of this cytokine was higher by approximately 27% than AE (*p* < 0.05) and approximately 3,5-fold higher than BE (*p* < 0.001) (Figure 7d). The concentration of TGF-β1 immediately AE was also higher than BE (*p* < 0.001), and 20 min after SAUNA-REST, as in the case of IL-10, the concentration of this cytokine was statistically significantly higher than AE (*p* < 0.001) and almost 3-fold higher than BE (*p* < 0.001) (Figure 7f). There were no statistically significant changes in the concentration of IL-1β and TNF-α (*p* > 0.05) after SAUNA-REST (Figure 7a,e). Moreover statistically significantly higher levels of IL-10 (*p* < 0.001) and TGF-β1 (*p* < 0.001) were observed after SAUNA-REST versus RT-REST (Figure 7d,f). However, no statistically significant differences in the concentrations of IL-1β, IL-6, IL-8 and TNF-α were observed (Figure 7a–c,e).

The concentration of IL-6 (*p* < 0.001) and TGF-β1 (*p* < 0.001) was higher after SAUNA-REST versus after CWI-REST (Figure 7b,f). On the other hand, the concentration of IL-8 (*p* < 0.05) was lower after SAUNA-REST than after CWI-REST (Figure 7c). There were no statistically significant differences in the concentrations of IL-1β, IL-10 and TNF-α comparing SAUNA-REST and CWI-REST (Figure 7a,d,e).

### 3.4. The Concentration of Cytokines in Novices

In the first stage of Experiment I, a statistically significant higher concentration of IL-1β (*p* < 0.05) and IL-6 (*p* < 0.001) in the blood serum of novices AE than BE was observed. 20 min after RT-REST the concentration of IL-1β was 2-fold lower than AE (*p* < 0.001), while the concentration of IL-6 20 min after RT-REST was still statistically significantly higher than BE (*p* < 0.001). A statistically significant increase of IL-10 concentration 20 min after RT-REST compared to the concentration BE and AE was found (*p* < 0.001) (Figure 8d). The concentration of TNF-decreased by about 26% AE (*p* < 0.05) versus BE. 20 min after RT-REST, the concentration of this cytokine increased by about 42% (*p* < 0.05) versus AE (Figure 8e). In this study, no statistically significant changes in the concentration of TGF-β1 and IL-8 AE and after RT-REST were observed (*p* > 0.05) (Figure 8c,f). In the second stage of the Experiment I the concentration of IL-1β 20 min after CWI-REST was statistically significantly lower (*p* < 0.05) than AE (Figure 8a). The concentration of IL-6 after CWI-REST was statistically significantly higher (*p* < 0.05) compared to AE (Figure 8b). There were no statistically significant changes in the concentration of IL-8, IL-10, TNF-α and TGF-β1 AE and after CWI-REST compared to the values BE (*p* > 0.05) (Figure 5c–f). Comparing RT-REST and CWI-REST, an approximately 31% lower IL-8 concentration (*p* < 0.05) was observed 20 min after CWI-REST compared to the concentration of this cytokine 20 min after RT-REST (Figure 8c). Moreover, the concentration of IL-10 (*p* < 0.001) was higher by about 23% after CWI-REST than after RT-REST (Figure 8d). However, no statistically significant differences were observed in the concentrations of IL-1β, IL-6, TNF-α and TGF-β1 (*p* > 0.05) (Figure 8a,b,e,f) comparing RT-REST and CWI-REST.

In the first stage of the Experiment II more than a 2-fold increase in the concentration of TGF-β1 (*p* < 0.001) was noted AE versus BE. 20 min after RT-REST the concentration of this cytokine was still statistically significantly higher (*p* < 0.001) than BE (Figure 9f). However, no statistically significant changes in the concentration of IL-1β, IL-6, IL-8, IL-10 and TNF-α (*p* > 0.05) were observed, both AE and RT-REST (Figure 9a–e). In the second stage of Experiment II, the concentration of IL-6 AE was about 4-fold higher than before exercise (*p* < 0.001), while 20 min after SAUNA-REST, it was even almost 6-fold higher than BE. Moreover, 20 min after SAUNA-REST a statistically significant increase in the concentration of this cytokine was observed (*p* < 0.001) compared to AE (Figure 9b). The concentration of TGF-β1 AE was about 85% higher than BE (*p* < 0.001), while 20 min after SAUNA-REST it was even almost 2.5-fold higher than BE (*p* < 0.001). Moreover, 20 min after SAUNA-REST, a statistically significant increase in the concentration of this cytokine by about 45% (*p* < 0.001) was observed compared to AE (Figure 9f). There was also a statistically significant increase in the concentration of IL-10 20 min after SAUNA-REST (*p* < 0.001) compared to AE. Moreover, the concentration of this cytokine was approximately 5-fold higher 20 min after SAUNA-REST than BE (*p* < 0.001) (Figure 9d). Similarly, there was a statistically significant increase in the concentration of TNF-α 20 min after SAUNA-REST (*p* < 0.001) compared to AE. The concentration of TNF-α was also higher by approximately 63% 20 min after SAUNA-REST than BE (*p* < 0.001) (Figure 9e). Comparing the concentration of selected cytokines after RT-REST and after SAUNA-REST, statistically significantly higher concentrations of IL-6 (*p* < 0.001), IL-10 (*p* < 0.001) and TGF-β1 (*p* < 0.05) were observed 20 min after SAUNA-REST (Figure 5b,d,f). However, no statistically significant differences in the concentration of IL-1β, IL-8 and TNF-α were observed after SAUNA-REST comparing to RT-REST (Figure 9a,c,e).

Statistically significant differences in the concentration of selected cytokines between SAUNA-REST and CWI-REST were noted. The concentration of IL-6 (*p* < 0.001), IL-8 (*p* < 0.001), TNF-α (*p* < 0.05) and TGF-β1 (*p* < 0.001) were higher 20 min after SAUNA-REST compared to the values obtained 20 min after CWI-REST (Figure 9b,c,e,f). However, no statistically significant differences were observed in the IL-1β and IL-10 concentrations (Figure 9a,d).

## 4. Discussion

The reduction of post-exercise muscle damage can protect the health and physical integrity of athletes, increasing their chances of achieving goals after a training cycle [48]. Due to a lack of scientific evidence, the selection of a form of post-exercise regeneration appropriate for given athlete is a serious dilemma of athletes, coaches and scientists involved in sports medicine [49].

### 4.1. The Influence of Cold Water Immersion on the Indicators of Inflammation and Oxidative Damage in Blood

Literature data suggest that exposure to cold may aid recovery by alleviating exercise-induced inflammation [16,50]. CWI has been shown to improve the well-being of athletes and to reduce swelling and muscle soreness; however, this mechanism has not been well described so far [20,51,52]. Only a few research studies have reported anti-inflammatory effect of CWI, while there have been many studies on the cytokine response to exercise [50,53]. In this study, the IL-6 level was higher after RT-REST and CWI-REST than before and after exercise. This result is in agreement with the findings of Roberts et al. [24], who showed increased IL-6 after CWI as well as after active regeneration (10-min cycling with low exercise intensity) after exercise. The increase in IL-6 expression after prolonged exposure to cold was also demonstrated by Rhind et al. [25]. Accordingly, Peake et al. [54] observed that the concentration of IL-6 was significantly higher after a CWI session than after exercise. On the other hand, White et al. [55] observed that CWI used after a high-intensity sprint run did not significantly reduce plasma concentrations of IL-6 and IL-8. Increased plasma levels of IL-6 may point to the sustained release of IL-6 from skeletal muscle in response to CWI-stimulated glycogenolysis [8,24]. The higher concentration of IL-6 after CWI-REST compared to the value AE in novices may result from modulation of muscle tissue as a consequence of exposure to cold. Different results obtained in this study and in some research conducted by other authors may result from differences in the status of participants subjected to exercise. Circulating levels of inflammatory markers are elevated with total and abdominal obesity, possibly owing to a higher secretion rate of cytokines by adipose tissue in obese people [56]. Lifestyle-related behavioral interventions, including changes in food intake/diet and regular physical activity, may have clinically significant benefits in improving responses to inflammation [57].

In presented study, the concentration of IL-10 was statistically significantly higher after CWI-REST than after RT-REST in both studied groups. IL-10 is believed to be the primary anti-inflammatory agent, as it inhibits the production of pro-inflammatory cytokines by activated monocytes and macrophages [8,58]. The concentration of this cytokine in winter swimmers was significantly higher after CWI-REST than AE and BE. A similar direction of changes in the concentration of IL-10 in the blood serum was observed in novices. These results are consistent with the findings of Bartley et al. [59], who showed an increase in the concentration of IL-10 after a 12-min CWI in competitors participating in the world triathlon championships. The results obtained in this study seem to confirm the positive effect of a single CWI as a factor regulating the local inflammation caused by exercise. Considering IL-10 concentration, the changes were comparable, while IL-6 concentration changed adversely in winter swimmers and novices, as the concentration of IL-6 in regular winter swimmers decreased both after RT-REST and CWI-REST.

In this study, no statistically significant changes in the TNF-α concentration were found in winter swimmers. This result might reflect the changes in the IL-6 levels, because TNF-α concentration depends on the secretion of IL-6, as this cytokine inhibits the expression of TNF-α [60]. It is postulated that prolonged immersion in cold water may adversely affect the regeneration process by inducing an increase in the concentration of TNF-α, lymphocytes and monocytes [61]. TNF-α has been shown to be a potent stimulator of muscle proteolysis in vivo [62]. Thus, an increase in TNF-α concentration may inhibit post-exercise regeneration. According to the literature data, the immunostimulatory effect of low temperature may be associated with the intensification of the noradrenaline response to cold [63]. On the other hand, published data suggest that repeated short-term exposure to cold has beneficial effects by inhibiting inflammation and mobilizing the immune system [63]. There was a delayed (after 6 and 12 h) decrease in TNF-α concentration after 10-min. CWI and after 170-min. intermittent CWI in young, healthy men not adapted to cold, but in these reported studies, exposure to cold was not preceded by exercise. In our study no changes in TNF-α in winter swimmers were observed, but IL-6 differed in athletes that are adapted and not adapted to cold. Rhind et al. [25] showed that exposure to cold after intense exercise resulted in a significant reduction in TNF-α concentration. They suggest that cold-induced changes in TNF-α expression could be associated with increased catecholamine secretion accompanying exposure to cold. In people not adapted to cold, CWI causes a reaction known as “cold shock” at water temperatures below 15 °C [64]. This physiological response is manifested by elevated levels of stress hormones including cortisol, adrenaline, and noradrenaline [8,64,65]. In the early stages of cold stress, glucocorticoids and catecholamines stimulate leukocytes to leave the organs and enter the blood and lymph vessels [66]. Then, due to the reflex of sympathetic activity, exposure to cold causes blood vessels to constrict. This, in turn, reduces the perfusion of loaded tissues, thereby reducing the circulatory exposure of the tissues to inflammatory cells [5]. CWI may therefore benefit recovery from exercise by inducing vasoconstriction and reducing infiltration of inflammatory cells into the muscles. According to this mechanism, CWI can reduce clinical signs of inflammation, including swelling of the limbs after exercise [67]. However, it is difficult to capture the ideal moment when the concentration of cytokines is significantly changed under the influence of the factors described above. It is possible that the changes occur at a different time point from the time of blood collection determined in this study. In this study, in novices, the concentration of TNF-α increased significantly to the values observed at the beginning of the study after 20 min of the rest, both at room temperature and including CWI. Both the blood sampling time and the decreased TNF-α concentration after submaximal exercise in novices may explain these discrepancies in the study results. It can be assumed that CWI did not lower TNF-α levels, because the load during exercise was not sufficient to promote an increase in the level of this inflammatory marker. The lack of statistically significant changes in the concentration of TNF-α may also be related to the changes in the concentration of IL-6 shown in this study.

In the case of IL-8, it was noticed that the concentration of this cytokine after CWI-REST was statistically significantly lower than after RT-REST. Contrary to the results obtained in this study, Bartley et al. [59] observed an increase in the concentration of IL-8 after a 12-min CWI in competitors participating in the world triathlon championships. This pro-inflammatory cytokine is released in the early phase of the inflammatory response [68]. Researchers postulate that CWI protocols with too long exposure to cold temperatures may exacerbate post-exercise inflammation.

Similarly to IL-8, a decrease in the concentration of IL-1β was observed in novices both after RT-REST and CWI-REST. The physiological stress induced by CWI increases tissue metabolism and appears to stimulate the immune system in volunteers participating in the study. IL-1β, a key pro-inflammatory cytokine, is considered a biomarker of systemic inflammation as well as of the stress response [69,70]. Thus, the results obtained in this study seem to partially confirm the beneficial effect of regular winter baths (which is the case in the winter swimmers group) on inflammation. At the same time, these results may suggest a possible role of this activity in the development of the functional plasticity of the organism’s stress response. TGF-β1 is another cytokine that is extremely important in the organism’s response to exercise, because it is involved in the process of muscle regeneration [71]. In this study, no statistically significant changes in the concentration of this cytokine in both winter swimmers and novices were observed as a result of the CWI.

Considering the activity of AAT and lysosomal enzymes, no statistically significant changes were observed after the use of various methods of post-exercise regeneration in both winter swimmers and novices. This could mean that short-term CWI used as a postexercise regeneration method did not damage the lysosomal membranes and lysosomal enzymes did not enter the bloodstream. It has been proven that lysosomal enzymes are involved in the formation of post-exercise damage to muscle cells [72,73]. Exercise causes greater oxygen consumption by the mitochondria, leading to increased formation of ROS, which results in peroxidation of the muscle cell membrane. Structural changes in muscle fibers caused by oxidative stress are accompanied by increased release of lysosomal hydrolases [4,74]. According to the results obtained in this study, Sutkowy et al. [75] observed that neither rest at room temperature nor rest combined with CWI affected the activity of lysosomal enzymes in the blood serum. Moreover, Mila-Kierzenkowska et al. [76] observed no changes in the activity of lysosomal enzymes in healthy participants after immersion in a river with a water temperature of 0 °C. CWI in water below 15 °C is physiologically stressful and can be dangerous to the organism [8,65]. The lack of changes in the activity of lysosomal enzymes in the blood serum of the participants of this study may suggest that the protocol of using a 3-min CWI is safe, as it did not damage the lysosomal membrane and release enzymes into the bloodstream. Similar results were reported by Mila-Kierzenkowska et al. [76], who studied the effect of cold water bathing on the activity of lysosomal enzymes in men who regularly undergo winter baths and in volunteers who had not previously used winter baths.

### 4.2. The Influence of Sauna Bath on the Indicators of Inflammation and Oxidative Damage in Blood

The reduction in post-exercise inflammatory response after a single session of exercise and CWI can be interpreted as a positive effect of exposure to low temperatures. However, it is well known that inflammation is an important part of the muscle repair process and that its reduction can be counterproductive in long-term training adaptations. Consequently, CWI appears to be a beneficial regenerative treatment, but only used in the short term. Long-term use of a cold water bath may not be beneficial in the process of adaptation to training. It is believed that “periodization of recovery” should be an important element in sports training programs [77].

There are numerous studies that have documented the positive effects of sauna on human health and overall efficiency of athletes [17,78,79,80]. However, exposure to heat also poses a challenge for the organism, which responds naturally to changes in ambient temperature [81]. Long-term hyperthermia may be a limiting factor in physical performance [82]. During exposure to heat stress, there is a reaction from the cardiovascular system aimed primarily at increasing blood flow through the skin, which is necessary for the proper dissipation of internal heat shock [34]. This response occurs through dilation of cutaneous blood vessels in combination with increased cardiac output and redistribution of blood flow and volume away from the central vascular bed, e.g., through visceral and renal circulation [83]. It is well known that as a result of overheating or exercise, genes encoding heat shock proteins (HSPs) and interleukins are induced [34]. In the available literature, there are only few reports on changes in the concentration of interleukins during a single Finnish sauna treatment, both in people who train regularly and in people who do not regularly perform exercise [34]. Occasional reports of the influence of the sauna on the organism of athletes concern changes in the number of leukocytes and the profile of white blood cells [31]. In this study, it was observed that the inflammatory response after a sauna bath is more intensified in novices than in the group of winter swimmers. This group showed higher concentrations of IL-6, IL-10 and TGF-β1 in the blood serum after SAUNA-REST than after RT-REST On the other hand, winter swimmers showed higher concentrations of IL-10 and TGF-β1 after SAUNA-REST than after RT-REST.

In this study, the concentration of IL-6 in novices after SAUNA-REST was statistically significantly higher than after RT-REST. There are studies showing that passive heating of the organism can cause a sharp increase in circulating IL-6 levels in young, healthy men, in overweight young men, and in men with a spinal cord injury [84,85,86,87]. However, during these tests, a prolonged (60–120 min) hot water (39–40 °C) immersion was used. In turn, Dugué and Leppänen [88], observed a significant increase in cortisol and IL-6 levels after a single sauna treatment. Similar results were obtained by Brenner et al. [62] in the experiment involving seven volunteers after a 1-h water bath at 35 or 38 °C, and then cooling the body in a climate chamber at 5 °C. Behzadi et al. [89] subjected middle-aged people to two 10-min sauna treatments, which resulted in a strong increase in IL-6 levels. The authors concluded that the increase in IL-6 is correlated with the intensity of the thermal stimulus. These results are consistent with the results obtained in this study; however, in the studies cited above, the sauna treatment was not preceded by physical exertion.

In this study, a statistically significant increase in the concentration of IL-10 in the serum of studied subjects after SAUNA-REST in comparison with the concentration of this cytokine AE was found. Moreover, in both groups a higher concentration of IL-10 was observed after SAUNA-REST than after RT-REST and these changes were more pronounced in the novices group. Considering the concentration of TNF-α, a statistically significant increase after SAUNA-REST versus AE and BE was found, while in winter swimmers no statistically significant changes in the concentration of TNF-α were observed. Moreover, analyzing the concentration of IL-8, no statistically significant changes were noted in novices as a result of using a sauna bath as a method of post-exercise regeneration. In turn, the concentration of IL-8 in winter swimmers was significantly reduced 20 min after SAUNA-REST compared to AE. Pilch et al. [90] investigated the effect of a sauna bath on the indicators of inflammation in trained and untrained men. The authors noted an increase in the concentration of IL-10 after completing a series of ten sauna treatments and before the last treatment only in the group of trained men. The researchers postulate that the adaptive increase in IL-10 was aimed at reducing the inflammatory response and the production of pro-inflammatory cytokines after the applied hyperthermia.

Moreover, in this study, an increase in the concentration of TGF-β1 after SAUNA-REST was noted in the novices as compared to BE. It was higher after a sauna bath than after RT-REST. The direction of changes in the concentration of this cytokine in winter swimmers was the same, but these changes were less marked than in novices. TGF-β1 plays a significant negative role during the muscle regeneration process. This cytokine is able to inhibit the proliferation of myoblasts as well as promote the formation of fibrosis. Thus, specific inhibition of the TGF-β1 signaling pathway can significantly improve the regenerative capacity of skeletal muscle [71,91].

The results obtained for IL-6, IL-8, IL-10, TNF-α and TGF-β1 may confirm the adaptive effect of regular winter baths on the adverse effects of thermal stress. A one-time sauna bath does not seem to be a significant stress factor for people who regularly use winter baths and may be a safe form of post-exercise regeneration for them. The disturbances of cellular homeostasis after a sauna bath are less harmful than for people who are not adapted to thermal stress. This fact can be explained by the occurrence of thermotolerance mechanisms in winter swimmers, thanks to which mild heat stress induces a state of temporary resistance to high temperatures [76]. Short-term but regular exposure to harmful conditions such as bathing in water during winter can lead to cellular adaptation and facilitate the maintenance of physiological cell functions under the influence of various stress factors [76]. Thermal stress can improve the readiness and effectiveness of the organism’s response to stress and the immune response by increasing the secretion of cytokines in people who regularly use winter baths [70]. The organism’s adaptation to repeated cold stress, previously postulated as an organism-hardening mechanism, may result in increased tolerance to stress and infectious diseases [76]. Dugue and Leppänen [88] found that the cytokine response after temperature stress (swimming in ice water combined with sauna treatment) was significantly higher in the group of people who did not regularly use winter baths than in the group of winter swimmers. The authors found that exposure to cold and heat affected both the endocrine and immune systems. Moreover, they indicated that the adaptation of individuals to changes in ambient temperature may occur when they are regularly subjected to stress. One-time heat treatments are a heavy burden on the organism, but repeated treatments alleviate the organism’s response to stress [90]. Exposure to high temperatures in a sauna immediately after training can enhance the organism’s adaptive response [92]. Using a sauna can also be a good option for physically inactive people to improve specific cellular responses and increase the production of anti-inflammatory cytokines [90]. It is worth emphasizing, however, that in order for these treatments to actually have a beneficial effect on the human organism and to support post-workout regeneration, they should be used as a series of treatments.

Sauna baths that combine heating and cooling of the human organism with a short effect of high air humidity [93], cause vasodilation and then contraction of all the body’s vessels [94]. The ischemia that occurs first, and then the organism congestion, may cause an increase in the production of ROS as a result of a reaction catalyzed by xanthine oxidase. Increased production of ROS can damage cell membranes [95]. The potential oxidative stress induced by the sauna treatment may damage lysosomal membranes and release lysosomal hydrolases into the peripheral blood [95]. It is believed that the source of the increased permeability of lysosomal membranes and the increase in the activity of lysosomal enzymes during or after exercise is an increase in pH in lysosomes and a decrease in the aggregation of enzymatic proteins inside them. These changes arise as a consequence of the accumulation of ammonia in the lysosomes, the concentration of which increases significantly in skeletal muscles and in the blood during exercise [96]. The effect of a sauna bath on the organism is considered to be similar to the effects of physical exercise; therefore, it is possible that ammonia accumulates in lysosomes also after this type of treatment [96]. In winter swimmers no statistically significant changes in the activity of AcP, ASA and CTS D were observed 20 min after SAUNA-REST compared to the results obtained AE and BE, while in novices the activity of AcP increases after SAUNA-REST and was higher than after RT-REST. In turn, the activity of CTS D in novices after the sauna bath was lower than AE. There were no statistically significant changes in AAT and ASA activity after the sauna treatment in novices, while in winter swimmers a statistically significant increase in AAT activity was noted 20 min after the sauna bath, compared to BE. Moreover, 20 min after RT-REST, AAT activity was statistically significantly lower than after SAUNA-REST. In their research, Mila-Kierzenkowska et al. [76] noticed that the sauna treatment significantly influences the activity of lysosomal enzymes and the activity of AAT both in winter swimmers and in novices. The authors observed that after the sauna bath CTS D activity decreased significantly, while ASA activity increased. In contrast, AcP activity remained unchanged in both groups. AAT activity increased following heat exposure in both experienced and novice winter swimmers. The increase in AAT activity in the blood serum due to the operation of the sauna can therefore be explained by the stress induced in the organism as a result of this treatment [96].

### 4.3. Comparison of the Influence of Cold Water Immersion and Sauna Bath as Methods of Post-Exercise Regeneration

Analyzing the results obtained in this study, it can be noticed that the concentration of selected cytokines (IL-6, IL-8 and TNF-α) in the blood serum of novices are higher after SAUNA-REST than after CWI-REST. Moreover, in the group of winter swimmers, only the concentration of IL-6 was higher, and the concentration of IL-8 was even lower after SAUNA-REST compared to CWI-REST. Therefore, it seems that CWI is a more effective method of one-time post-exercise regeneration used to reduce the inflammatory response induced by exercise compared to a sauna bath. Moreover, the concentration of TGF-β1 in both groups was higher after SAUNA-REST than after CWI-REST. This cytokine can inhibit myocyte proliferation [85]; therefore, the use of the biological regeneration method based on the use of low temperatures seems to have a more beneficial effect on the regeneration of skeletal muscles than a one-time sauna treatment. In this study, differences in the activity of AAT and lysosomal enzymes between the applied regeneration methods were also observed. The activity of AcP was higher, while the activity of CTS D was lower in the blood serum of the novices after SAUNA-REST compared to CWI-REST. Also, the activities of AAT, AcP and ASA in the serum of winter swimmer were higher after SAUNA-REST than after CWI-REST. It can therefore be concluded that the risk of proteolytic tissue damage is lower when using short-term CWI.

It should be emphasized that differences in the inflammatory response after CWI and after a sauna bath occurred probably not only due to the type of thermal shock, but also to different exposure times. The results presented in this study do not allow for an unambiguous comparison of the stress associated with exposure to cold and heat. The CWI session lasted only 3 min, while the sauna treatment lasted 23 min (two entries of 10 min and 3 min of rest). It is therefore possible that short-term exposure to cold was insufficient to induce changes in the inflammatory markers indicated. A longer stay in the conditions of high temperatures in the sauna favored the occurrence of changes in the tested parameters. However, a 20-min immersion in water at a temperature of 8 °C could be dangerous to the health of the study group. The CWI time used in this study was consistent with the standard protocols described in the available literature.

This study has some limitations. The studied groups were not homogenous. Some parameters describing these two groups differ significantly, including age, diet and BMI as well as the training status and cold adaptation. However, our intention was not to compare these groups directly, but only to demonstrate whether the nature of the changes in the examined parameters under the influence of CWI or a sauna bath was similar or different.

## 5. Conclusions

In conclusion, the influence of short-term CWI on the level of inflammatory markers during post-exercise regeneration turned out to be limited, but the use of a 3-min CWI seems to be harmless even for people who are not adapted to low ambient temperatures. It can be postulated that the modulation of cytokine concentration could occur much later than at the time points selected as the time of blood collection from the participants of this experiment. However, in the case of a single sauna bath, a greater effect on the parameters of inflammation and tissue damage was found, especially in people who are not regularly subjected to changed ambient temperatures. It seems that in order to obtain the beneficial effects of the sauna, a series of treatments should be used. Research should be continued, in which the CWI time, the number of sauna treatments or the time of blood collection after regeneration will be varied. This may allow for the development of safe and effective protocols of the use of CWI and a sauna bath as a means of supporting recovery after exercise.

## Figures and Tables

**Figure 1 antioxidants-11-02445-f001:**
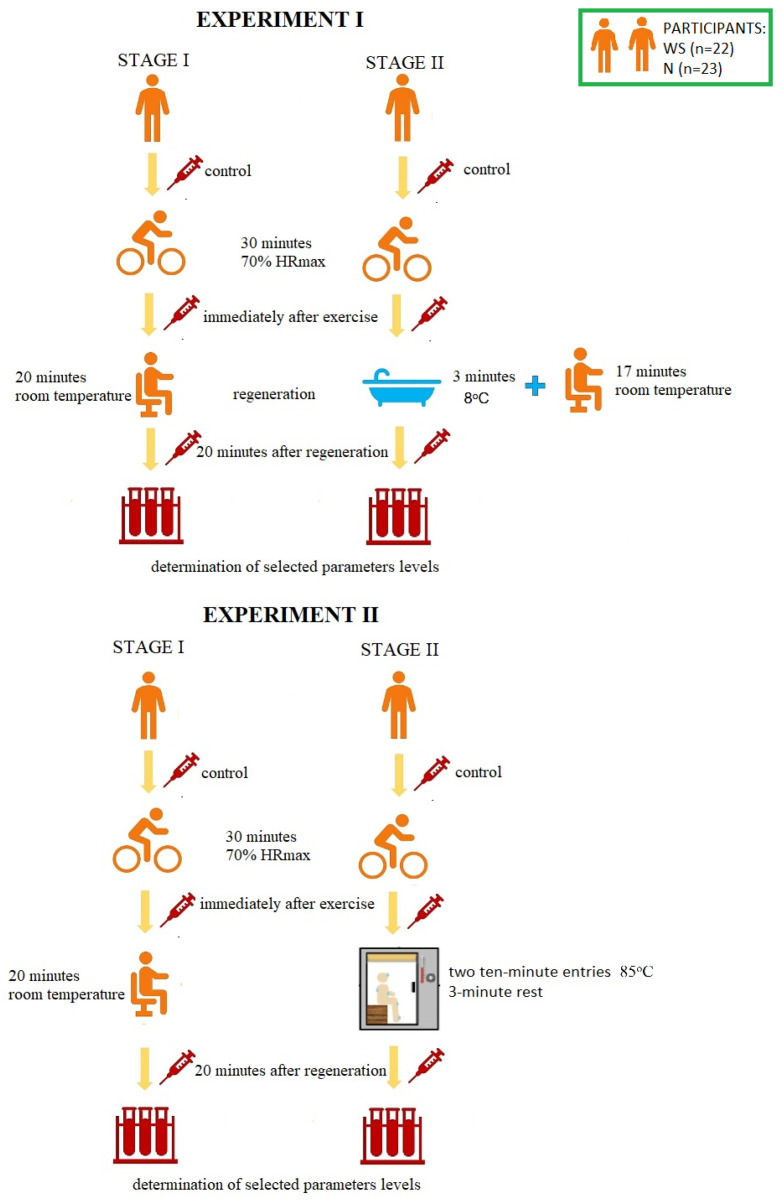
The scheme of the course of the research. HRmax—maximum heart rate, WS—winter swimmers, N—novices.

**Figure 2 antioxidants-11-02445-f002:**
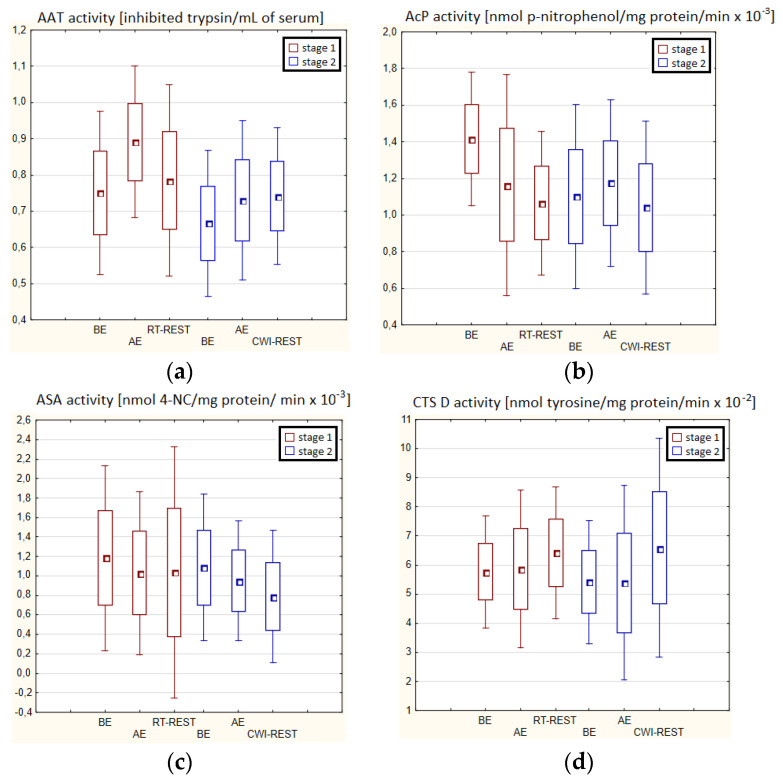
The activities of α1-antytrypsin and lysosomal enzymes in blood serum of winter swimmers (*n* = 22) in Experiment I: (**a**) AAT activity; (**b**) AcP activity; (**c**) ASA activity; (**d**) CTS D activity. Data are presented as the mean values ± SD. AAT—α1-antytrypsin, AcP—acid phosphatase, ASA—arylsulfatase, CTS D—cathepsin D, BE—before exercise, AF—after exercise, RT-REST—20-min recovery at room temperature, CWI-REST—20-min recovery at room temperature combined with 3-min cold water immersion.

**Figure 3 antioxidants-11-02445-f003:**
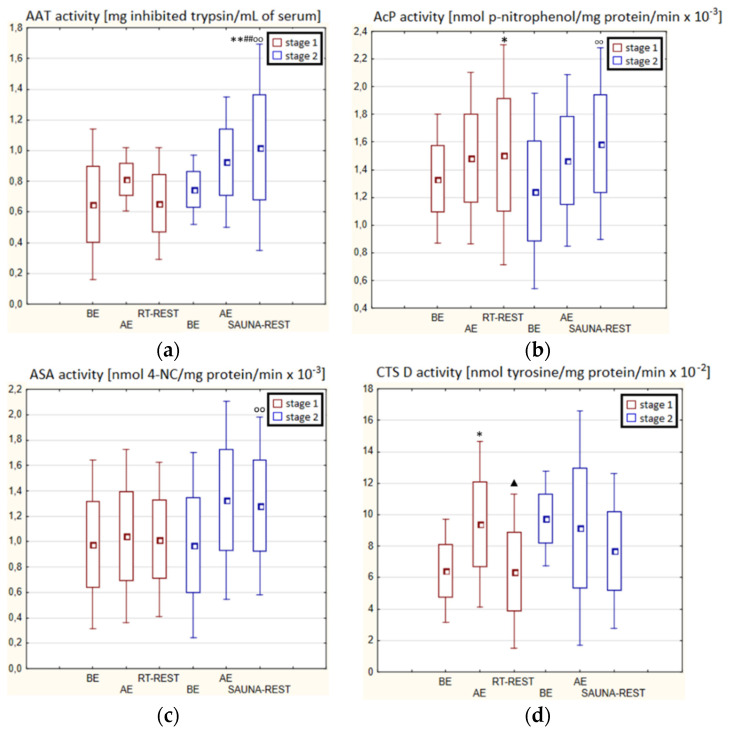
The activities of α1-antytrypsin and lysosomal enzymes in blood serum of winter swimmers (*n* = 22) in Experiment II: (**a**) AAT activity; (**b**) AcP activity; (**c**) ASA activity; (**d**) CTS D activity. Data are presented as the mean values ± SD. AAT—α1-antytrypsin, AcP—acid phosphatase, ASA—arylsulfatase, CTS D—cathepsin D, BE—before exercise, AF—after exercise, RT-REST—20-min recovery at room temperature, SAUNA-REST—rest combined with sauna bath, * *p* < 0.05 vs. BE, ** *p* < 0.001 vs. BE, ^▲^
*p* < 0.05 vs. AE, ^##^
*p* < 0.001 vs. RT-REST, ^oo^
*p* < 0.001 vs. CWI-REST.

**Figure 4 antioxidants-11-02445-f004:**
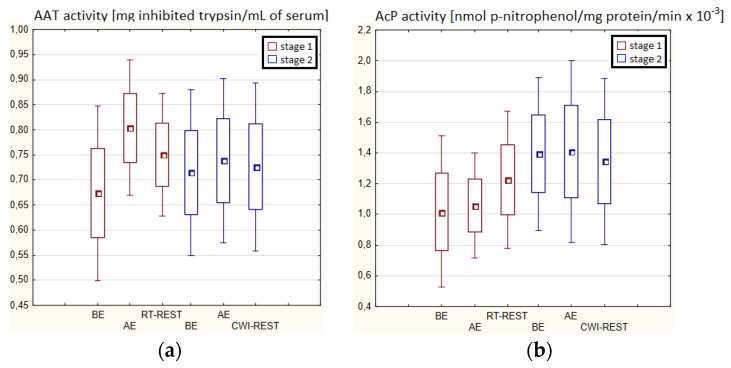
The activities of α1-antytrypsin and lysosomal enzymes in blood serum of novices (*n* = 23) in Experiment I: (**a**) AAT activity; (**b**) AcP activity; (**c**) ASA activity; (**d**) CTS D activity. Data are presented as the mean values ± SD. AAT—α1-antytrypsin, AcP—acid phosphatase, ASA—arylsulfatase, CTS D—cathepsin D, BE—before exercise, AF—after exercise, RT-REST—20-min recovery at room temperature, CWI-REST—20-min recovery at room temperature combined with 3-min cold water immersion.

**Figure 5 antioxidants-11-02445-f005:**
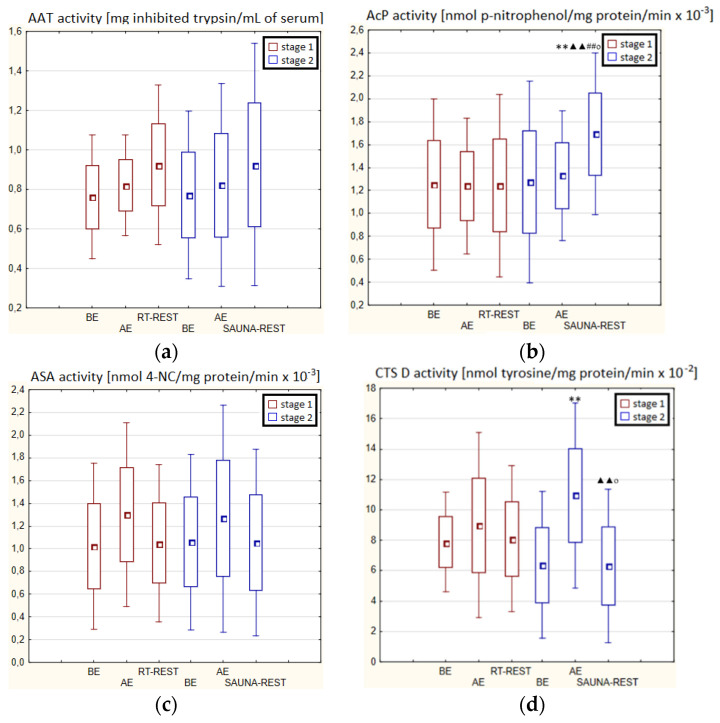
The activities of α1-antytrypsin and lysosomal enzymes in blood serum of novices (*n* = 23) in Experiment II: (**a**) AAT activity; (**b**) AcP activity; (**c**) ASA activity; (**d**) CTS D activity. Data are presented as the mean values ± SD. AAT—α1-antytrypsin, AcP—acid phosphatase, ASA—arylsulfatase, CTS D—cathepsin D, BE—before exercise, AF—after exercise, RT-REST—20-min recovery at room temperature, SAUNA-REST—rest combined with sauna bath, ** *p* < 0.001 vs. BE, ^▲▲^
*p* < 0.001 vs. AE, ^##^
*p* < 0.001 vs. RT-REST, ^o^
*p* < 0.05 vs. CWI-REST.

**Figure 6 antioxidants-11-02445-f006:**
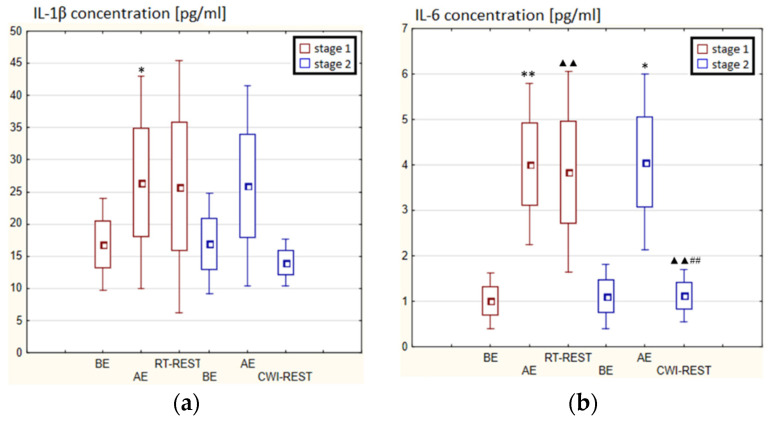
The concentration of selected cytokines in blood serum of winter swimmers (*n* = 22) in Experiment I: (**a**) IL-1β concentration; (**b**) IL-6 concentration; (**c**) IL-8 concentration; (**d**) IL-10 concentration; (**e**) TNF-α concentration; (**f**) TGF-β1 concentration. Data are presented as the mean values ± SD. IL-1β—interleukin 1β, IL-6—interleukin 6, IL-8—interleukin 8, IL-10—interleukin 10, TNF-α—tumor necrosis factor α, TGF-β1—transforming growth factor β1, BE—before exercise, AF—after exercise, RT-REST—20-min recovery at room temperature, CWI-REST—20-min recovery at room temperature combined with 3-min cold water immersion, * *p* < 0.05 vs. BE, ** *p* < 0.001 vs. BE, ^▲▲^
*p* < 0.001 vs. AE, ^##^
*p* < 0.001 vs. RT-REST.

**Figure 7 antioxidants-11-02445-f007:**
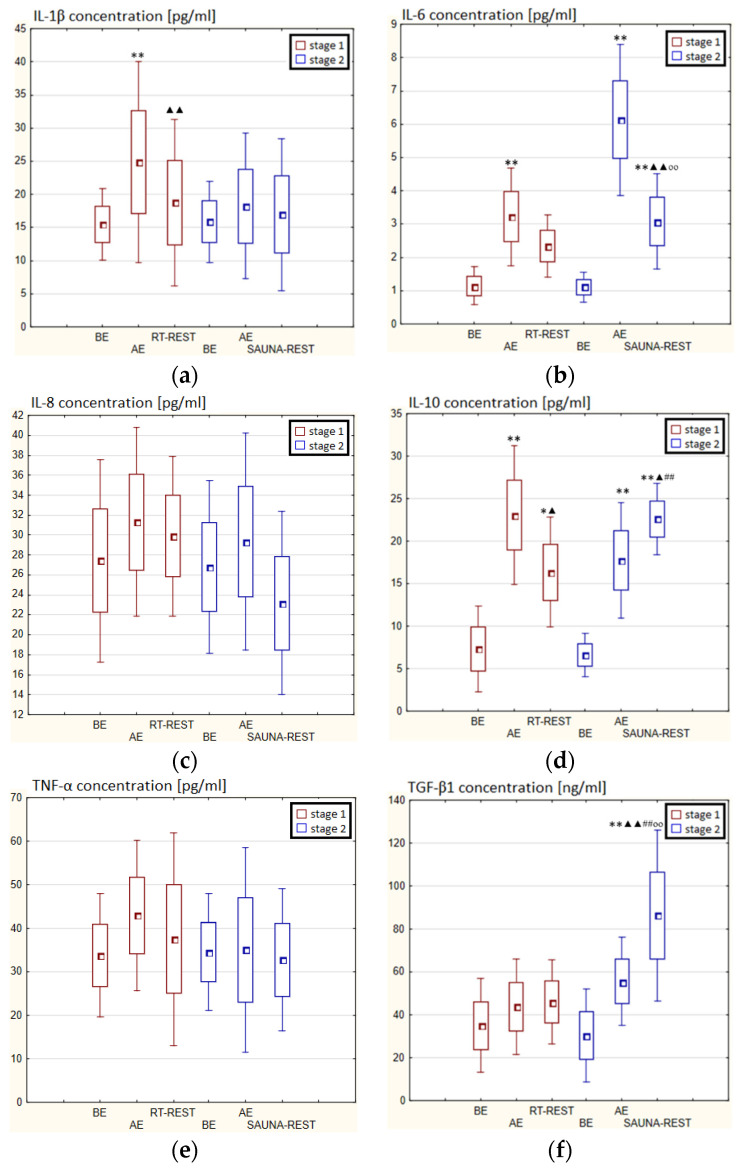
The concentration of selected cytokines in blood serum of winter swimmers (*n* = 22) in Experiment II: (**a**) IL-1β concentration; (**b**) IL-6 concentration; (**c**) IL-8 concentration; (**d**) IL-10 concentration; (**e**) TNF-α concentration; (**f**) TGF-β1 concentration. Data are presented as the mean values ± SD. IL-1β—interleukin 1β, IL-6—interleukin 6, IL-8—interleukin 8, IL-10—interleukin 10, TNF-α—tumor necrosis factor α, TGF-β1—transforming growth factor β1, BE—before exercise, AF—after exercise, RT-REST—20-min recovery at room temperature, SAUNA-REST—rest combined with sauna bath, * *p* < 0.05 vs. BE, ** *p* < 0.001 vs. BE, ^▲^
*p* < 0.05 vs. AE, ^▲▲^
*p* < 0.001 vs. AE, ^##^
*p* < 0.001 vs. RT-REST, ^oo^
*p* < 0.001 vs. CWI-REST.

**Figure 8 antioxidants-11-02445-f008:**
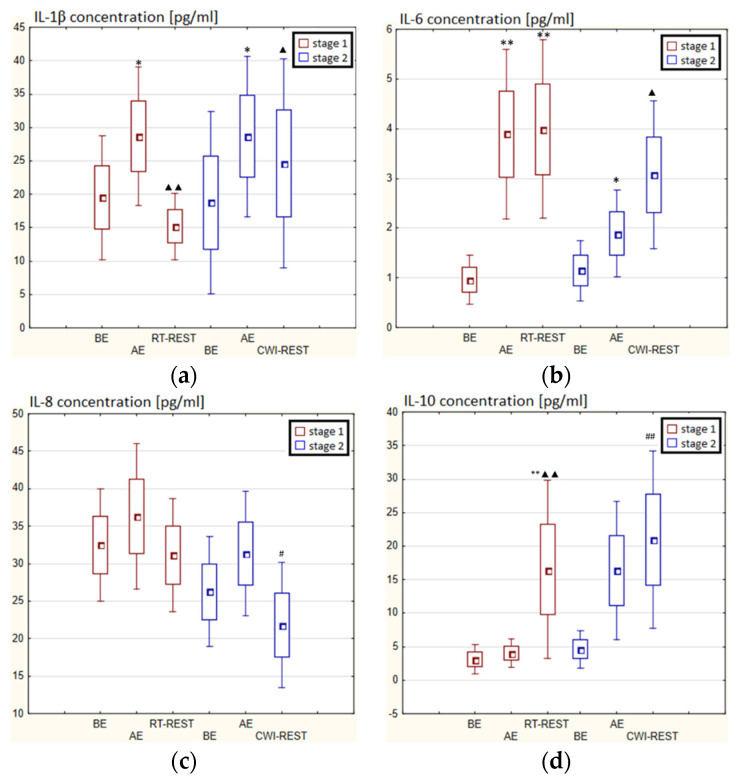
The concentration of selected cytokines in blood serum of novices (*n* = 23) in Experiment I: (**a**) IL-1β concentration; (**b**) IL-6 concentration; (**c**) IL-8 concentration; (**d**) IL-10 concentration; (**e**) TNF-α concentration; (**f**) TGF-β1 concentration. Data are presented as the mean values ± SD. IL-1β—interleukin 1β, IL-6—interleukin 6, IL-8—interleukin 8, IL-10—interleukin 10, TNF-α—tumor necrosis factor α, TGF-β1—transforming growth factor β1, BE—before exercise, AF—after exercise, RT-REST—20-min recovery at room temperature, CWI-REST—20-min recovery at room temperature combined with 3-min cold water immersion, * *p* < 0.05 vs. BE, ** *p* < 0.001 vs. BE, ^▲^
*p* < 0.05 vs. AE, ^▲▲^
*p* < 0.001 vs. AE, ^#^
*p* < 0.05 vs. RT-REST, ^##^
*p* < 0.001 vs. RT-REST.

**Figure 9 antioxidants-11-02445-f009:**
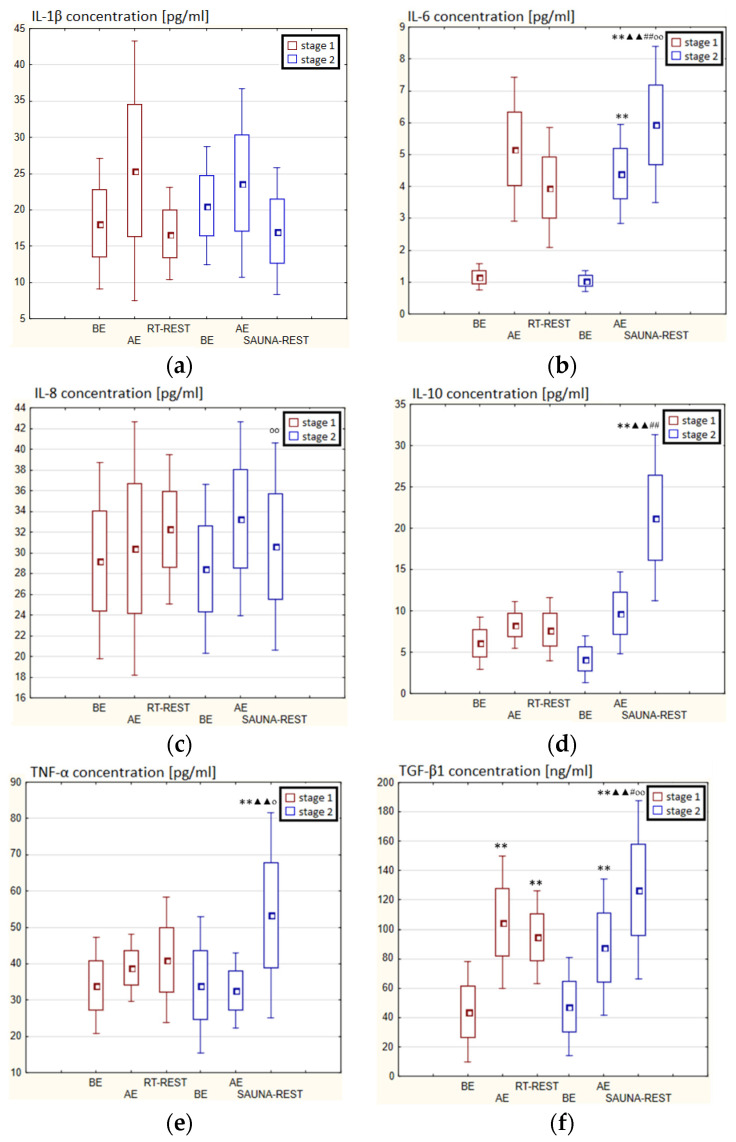
The concentration of selected cytokines in blood serum of novices (*n* = 23) in Experiment II: (**a**) IL-1β concentration; (**b**) IL-6 concentration; (**c**) IL-8 concentration; (**d**) IL-10 concentration; (**e**) TNF-α concentration; (**f**) TGF-β1 concentration. Data are presented as the mean values ± SD. IL-1β—interleukin 1β, IL-6—interleukin 6, IL-8—interleukin 8, IL-10—interleukin 10, TNF-α—tumor necrosis factor α, TGF-β1—transforming growth factor β1, BE—before exercise, AF—after exercise, RT-REST—20-min recovery at room temperature, SAUNA-REST—rest combined with sauna bath, * *p* < 0.05 vs. BE, ** *p* < 0.001 vs. BE, ^▲▲^
*p* < 0.001 vs. AE, ^#^
*p* < 0.05 vs. RT-REST, ^##^
*p* < 0.001 vs. RT-REST, ^o^
*p* < 0.05 vs. CWI-REST, ^oo^
*p* < 0.001 vs. CWI-REST.

**Table 1 antioxidants-11-02445-t001:** Basic characteristics of the study group (data are presented as means ± SD).

Parameter	Winter Swimmers	Novices
Group size [*n*]	22	23
Age [yr]	43.2 ± 5.9	25.0 ± 4.8
BH [cm]	174.6 ± 7.3	179.7 ± 5.0
BM [kg]	85.7 ± 13.9	81.4 ± 9.6
BMI [kg/m^2^]	28.06 ± 3.81	25.3 ± 2.7
BF [%]	22.5 ± 5.0	15.6 ± 4.3
TBW [%]	56.7 ± 3.7	61.5 ± 3.3
estimated VO_2max_ [mL/kg/min] ^1^	35.95 ± 6.6	40.95 ± 6.6
Borg CR10^1^	3.75 ± 0.91	4.06 ± 0.8
estimated VO_2max_ [mL/kg/min] ^2^	36.15 ± 7.4	40.67 ± 6.7
Borg CR10^2^	3.5 ± 1.2	4.08 ± 0.6
estimated VO_2max_ [mL/kg/min] ^3^	35.86 ± 7.2	40.85 ± 6.5
Borg CR10^3^	3.7 ± 1.1	4.07 ± 0.9
estimated VO_2max_ [mL/kg/min] ^4^	36.14 ± 6.9	40.87 ± 6.8
Borg CR10^4^	3.5 ± 0.9	4.02 ± 0.8
IPAQ	“moderate”—”high”	“moderate”—”high”

^1^ Experiment I Session I, ^2^ Experiment I Session II, ^3^ Experiment II Session I, ^4^ Experiment II Session II, BH—body height, BM—body mass, BF—body fat, TBW—total body water, VO_2max_—maximum oxygen consumption, Borg CR10—rating of perceived exertion scale.

## Data Availability

Data is contained within the article.

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
