# Peer review of "The Influence of Ambient Temperature Changes on the Indicators of Inflammation and Oxidative Damage in Blood after Submaximal Exercise"

_antioxidants, 2022, doi:10.3390/antiox11122445_

Round 1
Reviewer 1 Report
The work by Pawłowska et al. attempts to determine the effect of short-term cold-water immersion and sauna baths as methods of post-exercise regeneration on indicators of inflammation and oxidative damage in the blood of healthy recreational athletes. The controversial effects of both have been discussed, which makes this study timely. However, the authors do not provide sufficient evidence to support their conclusion. Some major and minor points need to be better explained or require editing, as follows:
Major points:
1. All figures need to be improved/changed: Please provide a graphic bar with an individualized presentation of each sample, not grouped (data points). Also, the data presentation is very confusing. It is not easy to read and understand the results. Please, reorganize the groups. I suggest you remove the duplicated BE and AE, once they are from the same methodological procedure. Then, in the graphic will be presented BE, AE, RT-Rest, and CWI/Sauna-Rest.
2. The data presentation is confusing. Sometimes you are comparing the Winter Swimmers x Novice. However, the samples present almost double the age difference and body fat percentage. Do you want to compare WS x N also? Please check these comparisons (Lines: 622-625; 763). How do control for these age differences?
3. Line 183: You mentioned you used the Tanaka formula to predict maximal heart rate. However, this formula has a 20% range of error. Did you check the maximal heart rate by some indirect method? Which method did you use?
4. About oxidative stress markers: Why didn’t you use other methods to measure oxidative stress? More results are necessary to infer some alteration in the systemic redox markers of this study. I strongly suggest you analyze: 8-Isoprostrane (lipid peroxidation); SOD activity; TBARS, and 8-Hydroxy-2'-deoxyguanosine.
5. Exercise capacity and -intensity measurements were not well addressed: Since you used the Tanaka formula to predict the maximal heart rate, and this formula has a range of 20% of the variation. Assuming 170 bpm as the max heart rate, you would have ± 34 bpm (136 to 204). Why did you use the Tanaka formula and not perform a submaximal exercise test and measure the maximal HR? Why did you calculate the exercise intensity based on the % of max HR and not on % of reserve HR (closer to VO2)? Did they keep at 70% of the max HR during the whole session? Did you use some range (e.g., 50% - 70% of maxHR)?
6. The WS group is almost twice the oldest as the Nov. If heart rate decreases according to age, and in the "170 Physical Fitness Test" you described that both should reach 170 bpm after the test. Do you believe the oldest group performed a higher intensity test related to novices?
7. Please proofread and double-check the English.
Minor points:
8. The introduction was very well organized and descriptive. However, I recommend you improve the explanation about ROS, exercise, and inflammation. ROS have been shown as molecules related to muscle adaptation response, not only related to damage. I suggest you add one sentence explaining the physiological roles of ROS in exercise, the difference between eustress and distress in redox biology and exercise adaptation field, and then explain about highly repetitive and intensive exercise sessions inducing muscle damage (please check 46-57, it is not only intensity).
9. Participants: If you want to compare WS x Nov, your groups should not have almost double the age. Also, the body fat percentage and VO2 appears to be different between the group. Did you do a statistical analysis of this data? Could you indicate it in the table?
10. Study design: It is not clear. How were the participants divided? Did they perform both experiments at different times? Did you subdivide participants? Did you ask if they are smokers? Did you give some orientations before the bioelectric impedance analysis?
11. Figure 1: Well designed, but some points need to be improved. Please, indicate the number of participants in each group and the abbreviations according to the figures.
12. What was the water temperature in CWI?
13. Table 1 – It was not indicated the estimated VO2. Since VO2 was not addressed by an indirect method, such as indirect calorimetry or ergospirometric effort test. Please, change to “estimated VO2”
14. Discussion: It is speculative and repeats what was written in the results topic. Beware of “in this study”. It is repetitive.
15. To finish, Lines 228-230 – The author mentioned the study aims to determine the effect of short-term cold-water immersion and sauna baths on the indicators of inflammation and oxidative damage. Nevertheless, the results are not addressing it.
Reviewer 2 Report
EVALUATION:
“The Influence of Ambient Temperature Changes on the Indicators of Inflammation and Oxidative Damage in Blood after Submaximal Exercise” by Pawlowska et al, submitted to ANTIOXIDANTS.
This manuscript studies 2 post-exercise regeneration systems: immersion in cold water for 3 min and sauna session. The results seem to indicate that cold water immersion is effective in reducing post-exercise inflammatory damage in trained (winter swimmers) as well as in non trained individuals (novice). Overall, the study gives interesting results that are largely discussed.
Nevertheless, there are some points that might be addressed by the authors in order to improve the final quality of the manuscript.
- ABSTRACT: IL-8 is repeated twice in L-31.
- INTRODUCTION: Authors indicate that the presence of lysosomal enzymes in serum is indicator of oxidative stress and only one reference is provided (Ref 7). Reference 8 is not in english and it is impossible to verify the information provided. First of all, lysosomal enzyme release seems to be more related to inflammatory damage that can derive later to an oxidative stress situation. In this context, authors have clearly focused in inflammation but not in oxidative stress, because markers of oxidative stress have not determined, such as malondialdehyde or protein carbonyls. Therefore, these aspects have to be taken into account in the INTRODUCTION.
- INTRODUCTION: Myeloperoxidase is a main marker during inflammatory processes and has been used in many reports to study post-exercise recovery. Why the authors have not chosen this marker? This needs an explanation in this section.
- INTRODUCTION: The final paragraph (L-10-110) should better describe the objectives of the study. It has several form errors, i.e. “there is a scientific reports…” (L-104).
- MATERIALS AND METHODS: In L-117 “eating habits are mentioned, but are they homogeneous between groups? Are they adapted to the sport discipline that participants are performing? Unbalanced nutrition is key factor in oxidative stress production. How this point has been controlled by researchers?
- MATERIALS AND METHODS and DISCUSSION: The equality of groups is mentioned in L-241. However, this assumption is not true. The groups are different is many aspects: likely the diet, the training status (winter swimmers vs novice), body fat (higher in winter swimmers), cold adaptation (better in winter swimmers). All these differences can condition the subsequent Discussion, but only the training status and cold adaptation have been addressed properly. See L-492-496, L-564-566, L-622-624. Authors should mention in some part of Discussion the possible role of diet and body composition in the inflammatory parameters studied. They might indicate as well the differences between groups and avoid comparisons between them, because they are not homogeneous.
- RESULTS: Taking into account the differences between groups, the analysis has to be focussed in intragroup differences during the different phases of the intervention. Comparisons between groups should be mentioned with caution, advertising that groups are different and interpretation of results has to take into account these differences. Nevertheless, the difference between groups could be considered as a limitation of the study that can be mentioned at the end of the Discussion section.
Round 2
Reviewer 1 Report
Thank you for taking my comments into consideration.